# Diagnosis and Treatment of Lower Extremity Arterial Disease—A Survey among Family Medicine Trainees in Poland

**DOI:** 10.3390/ijerph20021392

**Published:** 2023-01-12

**Authors:** Aleksandra Danieluk, Anna Niemcunowicz-Janica, Adam Windak, Sławomir Chlabicz

**Affiliations:** 1Department of Family Medicine, Medical University of Bialystok, 15-054 Bialystok, Poland; 2Department of Forensic Medicine, Medical University of Bialystok, 15-269 Bialystok, Poland; 3Department of Family Medicine, Collegium Medicum, Jagiellonian University, 31-061 Krakow, Poland

**Keywords:** ankle-brachial index, peripheral artery disease, lower extremity artery disease, family medicine

## Abstract

Guidelines point to the ankle-brachial index (ABI) as a non-invasive tool for the initial diagnosis of lower extremity artery disease (LEAD). Questions have been raised whether primary practices should perform ABI. An online questionnaire was distributed among family medicine trainees in two academic centers in Poland. The questionnaire aimed to establish their knowledge about LEAD management and their opinion on the usefulness of ABI measurement and other LEAD diagnostic methods in primary care. ABI measurement was found either very or moderately useful in LEAD diagnosis by 94.5% of the respondents. Among the three most important elements of LEAD management, lifestyle changes, secondary prevention of atherosclerosis and exercise treatment were chosen, respectively, by 98.6%, 83.6% and 72.6% of them. ABI was seen as a useful diagnostic method at the primary care by 74% of the participants; however, 82.2% of them do not have access to ABI measurement in their workplace. The residents have good knowledge of the diagnostic methods of LEAD and consider ABI measurement as useful in LEAD diagnosis. However, most of them do not have access to ABI measurements in their clinical practices. Future discussion and potential financial changes will be needed for the introduction of ABI measurements into Polish primary care.

## 1. Introduction

Peripheral artery disease (PAD) is an atherosclerotic disease that has an increasing impact on public health, with the estimated number of affected individuals in the world rising from 202 million in 2010 to over 236 million in 2015 [1,2]. The prevalence of the disease is higher in high-income compared to low-income countries and increases with age, reaching the level of 6.5% in men, 8.21% in women and 7.37% overall in individuals from high-income countries aged over 25 years [2]. PAD refers to atherosclerosis in any of the peripheral arteries leading to their dysfunction. Given that the lower extremities are the most common locality of the pathology, a new term, lower extremity artery disease (LEAD), was coined; in addition, separate guidelines referring only to that location of the disease have been published [3].

The typical symptoms of the disease include claudication, which is defined as exertional pain in the lower extremities relieved after rest. One of the most severe complications of LEAD is limb-threatening ischemia, which presents as gangrene, rest pain and ulcerations and can lead to limb loss or even patient death. What is more, the presence of LEAD is very often associated with arterial disease present in another arterial site. It has been found that among LEAD patients, 25–70% also suffer from coronary artery disease, 14–19% from carotid artery stenosis and 10–23% from renal artery stenosis [4]. However, about 40% of the patients with LEAD remain asymptomatic, which poses a great challenge in the diagnosis [5]. It was shown that the Edinburgh Claudication Questionnaire, which is a tool used for the identification of symptomatic LEAD individuals, has a 52.5% sensitivity and an 87.1% specificity in diagnosing the disease [6]. The low agreement between the symptoms and the presence of LEAD is especially notable in the diabetic population [7].

Introducing early diagnosis and proper treatment at the asymptomatic stage of the disease can prevent progression to the next stages [5]. Furthermore, even asymptomatic patients are burdened with a higher cardiovascular risk [8]; they should be observed more promptly and treated accordingly with the use of lifestyle interventions and, in many cases, pharmacological agents such as antiplatelet drugs and statins. Access to early diagnosis may also prevent misdiagnosis. Many of the patients with LEAD present with atypical symptoms, such as numbness, cramping and weakness in lower extremity muscles [9]. This can lead to incorrect diagnoses, such as chronic venous disease, vitamin deficiency or neuropathy, resulting in unnecessary and sometimes harmful treatments.

The current guidelines point to the ankle-brachial index (ABI) as a primary non-invasive tool for the initial diagnosis of LEAD [10]. ABI measurement is usually performed by measuring the patient’s blood pressure at the level of the arms and at the level of the calves, followed by dividing the highest of the obtained results at the calf level by the highest, or, in some cases, the mean result at the arm level. Results lower than 0.9 are considered abnormal and a result lower than 0.5 is considered critical ischemia [4]. The wound, ischemia, foot infection (WIfI) classification also defines critical ischemia as an ABI lower than 0.4, an ankle blood pressure lower than 50 mmHg, a toe blood pressure lower than 30 mmHg and a transcutaneous oxygen pressure lower than 30 mmHg [11]. The blood pressure measurement at the level of the calf is usually taken with the doppler probe. A study by Lange et al. determined that utilizing the highest arterial ankle pressure brings the most accurate results for LEAD diagnosis [12]. Resting ABI specificity is assessed at 83% to 96% in different studies and its sensitivity is at 61% to 73% [13]; however, other studies place it at 95% sensitivity and nearly 100% specificity [4]. Furthermore, a stress test following ABI measurement can increase disease detection by around 30% [4]. In addition to being a diagnostic tool for LEAD, studies have shown that an abnormal ABI result is connected to increased total mortality, cardiovascular mortality and coronary event risk. It has been suggested that ABI measurement can lead to reclassifying a patient’s cardiovascular risk category [14]. An epidemiological trial carried out in Germany (getABI study) has aimed to assess the difference in mortality from any cause as well as the risk of vascular events in groups in abnormal and normal ABI, and has shown ABI to be a strong predictor of stroke [15,16]. ABI is an inexpensive test, which can potentially be applied in general practice to assess a broad group of patients. However, previous research shows that primary care physicians often either do not perform ABI measurements or perform them infrequently or incorrectly [17,18,19,20]. Studies show that even though LEAD is a serious condition with a potentially severe impact on the patient’s health, many general practitioners do not employ all the means necessary to diagnose it and to prevent its further progression.

Questions have been raised about whether primary care physicians should perform ABI in their patients. There are certain doubts about the diagnostic accuracy of resting ABI and its underestimation of the severity of the disease [21]. Some studies suggest a duplex ultrasound or postexercise ABI as a further evaluation in patients with suspected LEAD and normal ABI results [13,21]. Additionally, studies show that a significant proportion of patients with suspected LEAD have incompressible arteries or have typical symptoms regardless of a normal ABI result, which makes the measurement less reliable [22]. Joined recommendations of the Polish Society for Vascular Surgery, Polish Society for Hypertension, Polish Society for Wound Management and Polish Society for Cardiology on LEAD management suggest that in most symptomatic LEAD patients, further evaluation with duplex ultrasound will be necessary regardless of the ABI result [23]. In the Polish healthcare system, such a recommendation warrants a specialist referral of all LEAD patients.

We present a survey study carried out among family medicine trainees in Poland. The survey aims to establish their current knowledge about the principles of LEAD management and their opinion on the usefulness of ABI measurement and other diagnostic methods for LEAD in primary care.

## 2. Materials and Methods

An online survey was conducted among family medicine trainees in two major academic centers in Poland: Białystok (Medical University of Białystok, Department of Family Medicine) and Kraków (Jagiellonian University, Department of Family Medicine) in March and April 2021. The survey consisted of twelve questions, out of which seven were single-select multiple choice questions, four were multiple-select multiple choice questions and one was a question about the year of graduation of the respondent. In the single-select multiple choice questions, the respondents were asked to establish whether the physical examination can exclude LEAD diagnosis, whether they have a possibility to perform ABI in their practices, whether they believe ABI should be performed in primary care offices and if they had a chance to observe ABI measurements during their university education. They were also asked to indicate how useful they find ABI measurement to be in symptomatic LEAD and who should perform ABI in primary care according to their opinion. In the multiple-select multiple choice questions, the respondents were asked about the methods of diagnosis of LEAD, indications to perform ABI other than LEAD, management of LEAD patients and barriers to ABI measurement in primary care. The survey was modeled to assess two elements: knowledge of the current guidelines on peripheral artery disease and the perceived clinical utility of ABI in primary care. The questionnaire was uploaded to an online survey platform and the link was mailed out to the respondents.

Data were analyzed using descriptive statistics.

The study gained Medical University of Bialystok Ethics Committee approval (approval number APK.002.433.2020).

## 3. Results

### 3.1. Professional Status of the Respondents

A total of 73 responses were recorded. The surveyed trainees graduated from medical school between the years 1988 and 2019, with a significant majority (72.6%) graduating after the year 2015. Considering the family medicine residency program in Poland, which includes hospital attachments when the trainees do not actively treat patients in primary care facilities, the respondents were asked whether they provide care for patients independently at the family doctor’s practice.

### 3.2. Views on Lower Extremity Artery Disease Diagnosis

Among the respondents, 94.5% indicated that the presence of a lower extremity pulse on physical examination does not exclude the possibility of LEAD—deeming physical examination not sufficient for LEAD diagnosis. Nearly all respondents—respectively, 94.5%, 94.5% and 93.2%—reported the patient’s history, physical examination and ABI measurement as necessary for LEAD diagnosis. Slightly over 60% of the trainees also perceived a doppler ultrasound as a required diagnostic tool. A specialist consultation is necessary to make a LEAD diagnosis according to 31.5% of participants (Figure 1).

The ABI measurement was found either very or moderately useful in LEAD diagnosis by 94.5% of the trainees. Three respondents reported that the utility of ABI in LEAD diagnosis is hard to assess and one respondent found it not useful (Figure 2).

Only slightly more than half of the trainees—54.8%—indicated that they had an opportunity to see the methodology of ABI measurement during their university education.

When asked about indications to perform ABI measurement other than LEAD suspicion, the majority of the trainees pointed out a patient assessment before compression therapy (58.9%), swelling in the lower extremities (58.9%), lower extremity ulcerations (76.7%) and diabetic foot disease (54.8%). Other possible indications chosen by the respondents were erysipelas (5.5%) and hypertension (1.4%).

### 3.3. Views on Lower Extremity Artery Disease Treatment

To determine the state of the trainees’ knowledge of LEAD treatment, the respondents were asked “What treatment for a LEAD patient should be introduced in primary care? Choose the 3 most important elements of the treatment”. The three most chosen answers were lifestyle changes (e.g., smoking cessation, diet, weight reduction), secondary prevention of cardiovascular events (e.g., statins and antiplatelets) and exercise treatment (e.g., walking), chosen, respectively, by 98.6%, 83.6% and 72.6% of the respondents. In total, 65.8% of the trainees indicated pharmacological treatment, such as pentoxiphylline and cilostazol, as one of the three answers chosen in this question.

### 3.4. Views on the Ankle-Brachial Index in Primary Care

The survey showed that 74% of the trainees see ABI as a diagnostic modality that should be performed at the primary care level (Figure 3), while 82.2% of the trainees do not have access to ABI measurement in their workplace (Figure 4). A lower number of respondents—8.2%—replied that ABI should not be performed at the primary care level. When indicating what staff should be responsible for ABI performance in patients, most trainees pointed to the primary care nurse (56.2%) or primary care doctor (31.5%), while 5.5% see it as a specialist care responsibility. Two respondents (2.7%) do not see the need of performing ABI measurements in patients.

As for the barriers to ABI performance, a huge majority of the respondents pointed to time restraints (80.2%) and lack of equipment (76.7%). A smaller number of trainees see staff shortage, lack of skill in performing or interpreting the measurement and lack of financial reimbursement for the measurement (respectively, 42.5%, 28.8%, 20.5% and 30.1%).

## 4. Discussion

### 4.1. Summary of Main Findings

The results of our survey clearly show that family medicine trainees in Poland have a good understanding of the recommended initial diagnostic method for peripheral artery disease. Most respondents saw the patient’s history and physical examination as insufficient to diagnose LEAD and pointed to the need for either an ABI measurement or a doppler ultrasound. That being said, even though the knowledge of the guidelines is correct, the trainee clinicians often have no access to ABI measurement equipment at their workplace and many of them did not learn ABI measurement techniques during university education; thus, the chance to incorporate the recommended initial diagnosis tool into their clinical practice is low. Without access to an ABI measurement, all the patients with LEAD suspicion need to be referred to specialized care, which results in a long time to diagnosis and delayed treatment.

Another puzzling finding is the fact that even though the majority of the participants found ABI useful, to some extent, in LEAD diagnosis, almost half of them only found it “moderately” useful. Further questions can be asked as to the potential cause for this result, which can be caused by their own insecurity in performing the measurement or concerns of the potential limitations of the test; for example, caused by arterial calcification or diabetes.

When it comes to limitations to ABI introduction, the largest number of respondents pointed to time restraints. According to the answers concerning LEAD treatment, the majority of the trainees correctly identified lifestyle changes, secondary prevention of atherosclerosis and exercise treatment as the most vital elements of LEAD treatment.

### 4.2. Comparison with Previous Literature

The results of our study show that a lower percentage of respondents perform ABI in their practices than in the previously performed studies. A study performed in France in 2021 showed that the majority of the surveyed general practitioners did not perform ABI in their practices, reporting only 6 respondents practicing ABI out of 92 survey participants [24]. A survey performed in Wales in 2014 reported that 27% of the responding general practitioners did not perform ABI measurements at their practices and referred the patients requiring the test to specialized care and 73% of them performed ABI measurements less often than four times per month [17]. In another survey, performed in 2013 among Australian general practitioners, 58% of the respondents reported never measuring ABI in their patients. However, in the same study, 70% of the respondents indicated that they used duplex ultrasound as the first-line LEAD diagnosis tool; so, it is potentially possible that good access to a duplex ultrasound deemed ABI measurements less useful in the everyday practice of the responding physicians [25]. Another study by McGuckin et al. reported that 27% of the practices had access to a doppler system for ABI measurement [26]. Furthermore, in a cross-sectional study by Hageman et al., which investigated the level of adherence to PAD management guidelines in patients referred by general practitioners with PAD suspicion, 42% of the patients with no previous PAD diagnosis did not undergo ABI measurements before the referral [20].

A study by Meyer et al. analyzing barriers to ABI measurement found the time needed to perform to be the main obstacle to the diagnostic method [19]. Similarly, in a study by Yap Kannan et al., time constraints were one of the most reported limitations, along with staff availability and staff training [27]. Those findings are similar to what we observed in our study. A study by Rochoy et al. reported routine referral to an angiology specialist among the barriers noted by general practitioners [24].

In a study by Hageman et al., which analyzed the adherence to the guidelines on PAD management, 55% of the patients with suspected PAD were prescribed statins or other lipid-lowering treatment and 59% were prescribed antiplatelet agents before being referred to secondary care [20]. A recent study performed among primary care providers in Saudi Arabia showed that 68.2% of them prescribed antiplatelet therapy and 13.2% of them knew lipid, blood pressure and glycemic target goals for PAD patients [28]. Assuming that the responding trainees stand by their answers on the LEAD treatment in clinical situations, a better level of adherence to treatment guidelines can be expected in newly diagnosed LEAD cases treated by the respondents.

### 4.3. Strengths and Limitations

The study suffers from potential limitations. First of all, it was performed among trainees from only two locations, leaving the possibility of different results in different geographical areas of the country. Second, it was distributed via email to trainees taking part in training courses, potentially not reaching all of the trainees in selected areas. The study sample is relatively small.

Among the study strengths, it is important to mention that the multiple-answer survey questions included an option to add the respondents’ own answer, which increased the probability of the respondents providing their actual opinions on the topics, rather than having to choose only from the pre-designed ones. Moreover, it is the first study of this type, with the aim of assessing opinions on LEAD diagnosis and treatment, performed in Poland. The condition on which the study concentrates is widely spread and often diagnosed and treated late, making the subject relevant for clinical practice.

### 4.4. Recommendations for Clinical Practice and Further Research

Current guidelines and the healthcare system situation in Poland put the responsibility for LEAD diagnosis on specialized care, leaving only the initial suspicion and specialist referral to the primary care clinicians. In specialized care, ABI measurements are accessible; however, most patients are also referred for further evaluation, usually with a doppler ultrasound. Usually, only patients with the most typical and significant symptoms are referred to specialist care. With a high number of patients with suspected vascular disease and relatively low access to specialist care, the patients with suspected LEAD often wait months for a definitive diagnosis and the patients with atypical symptoms often remain undiagnosed.

Introducing the possibility of ABI measurement in primary care could broaden the group of patients that are diagnosed, shorten the time needed for diagnosis and, thus, precipitate the introduction of the treatment. The survey presented above shows that the family medicine trainees have sufficient knowledge of the disease diagnosis and management to justify such a modification. A worrying finding, however, is the fact that almost half of the respondents did not observe an ABI measurement during their university education. A study by Nexøe et al. showed that after a short course in ABI measurements, an unacceptable number of false-positive results was present, which caused the authors to suggest either prolonged education in the matter or only performing ABI in a specialized setting [29]. If university-level education lacks even a short instruction in ABI measurement, it is unlikely for future clinicians to pursue prolonged education in the matter, which will result in leaving the ABI measurements solely to specialized care.

Currently, there is a lack of proper motivation for primary care clinics to diagnose LEAD, which influences the situation negatively. The trainees see the need for ABI accessibility in primary care and point to trained nurses as medical professionals who could be responsible for such measurements. Similar solutions have been introduced in other countries. The potential incentive in Poland could be financing ABI as a fee for service separate from the capitation fee. This type of financing is gradually being introduced to primary care in Poland in the case of other diagnostic modalities.

## 5. Conclusions

The results of our survey highlight the advantages and shortcomings of medical education on LEAD in Poland rather clearly. The trainees have good knowledge of the diagnostic methods of LEAD and consider ABI measurement as useful in LEAD diagnosis. However, the trainees may encounter problems when their theoretical knowledge crosses paths with clinical practice. Most of the respondents do not have access to ABI measuring equipment in their clinical practices. Many participants did not have practical knowledge on the ABI performance presented to them during university education. It is important to put more emphasis on the practical training in ABI measurements and awareness of its potential use.

Given that ABI sensitivity can be found insufficient to qualify it as a screening test, a discussion should be carried out to establish whether or not performing it in a broad primary care population is viable. However, if the measurements are to be implemented, changes in financing of the diagnostic methods in primary care might prove necessary to aid ABI measurement implementations at that level of healthcare in Poland.

## Figures and Tables

**Figure 1 ijerph-20-01392-f001:**
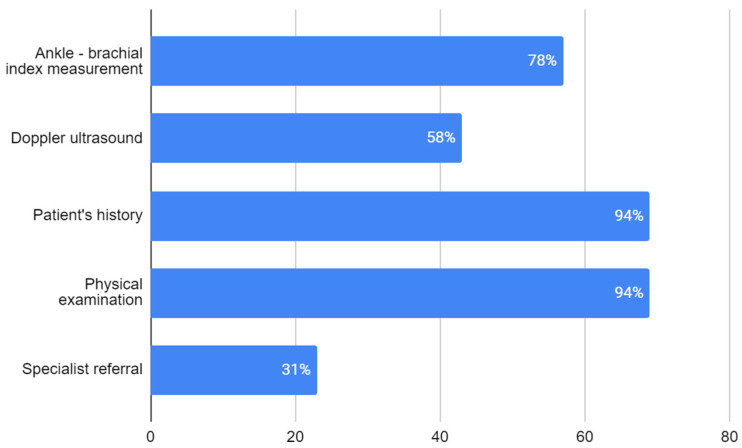
Respondents’ opinion on procedures necessary for diagnosis of LEAD (no. of responses). Percentage of respondents choosing each answer is given in white numbers.

**Figure 2 ijerph-20-01392-f002:**
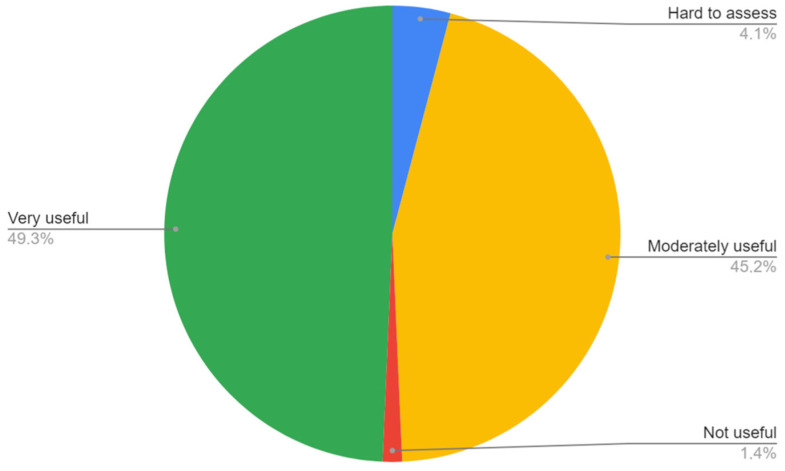
Respondents’ opinion on utility of ABI in diagnosis of LEAD.

**Figure 3 ijerph-20-01392-f003:**
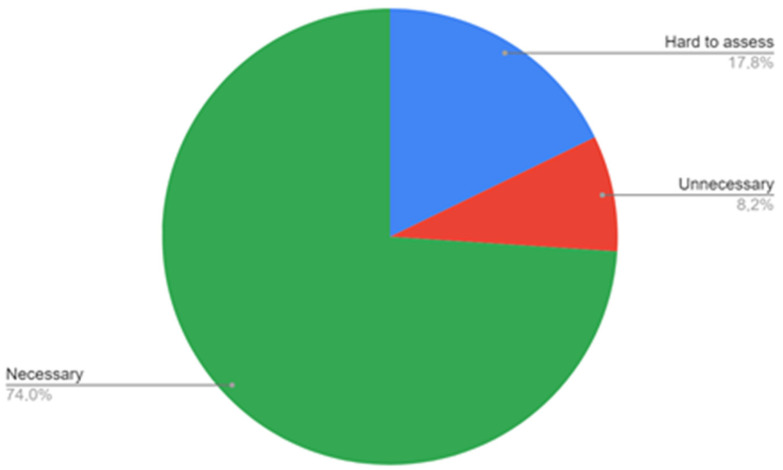
Respondents’ opinion on the necessity of ABI measurements in primary care.

**Figure 4 ijerph-20-01392-f004:**
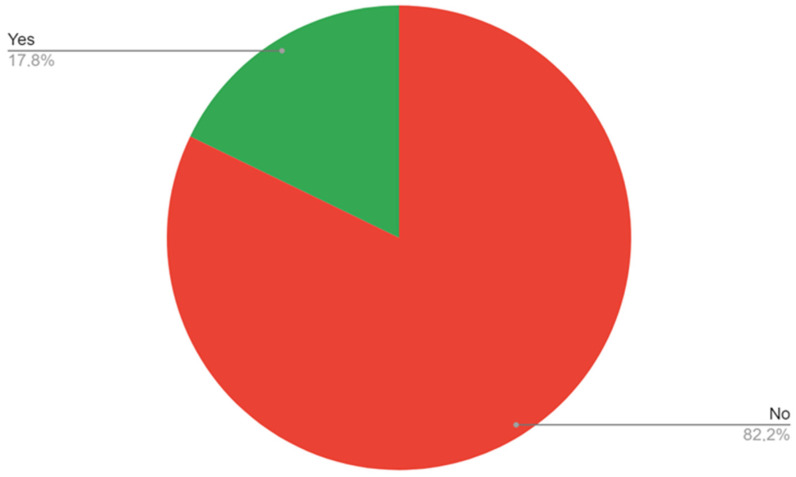
Access to ABI measurements in the respondents’ workplace.

## Data Availability

The data presented in this study are available on request from the corresponding author. The data are not publicly available due to privacy of respondents.

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
