# Peer review of "Diagnosis and Treatment of Lower Extremity Arterial Disease—A Survey among Family Medicine Trainees in Poland"

_ijerph, 2023, doi:10.3390/ijerph20021392_

Round 1

Reviewer 1 Report

-       Moderate english revision is needed (e.g. abstract: line 19. “a diagnostic useful” à “an useful diagnostic method”; Page 2 line 51: “antiplatelet factors” à “antiplatelet agents” or “antiplatelet drugs”)

-          Please do not start sentences with numbers (e.g. 3 à “Three”)

-          “Second of all” in line 213 à “Second”

-          Please specify better in the introduction the technique for measuring ABI and the evidence on its role in the diagnosis of LEAD

-          One hypothetical risk of the implementation of the ABI as a primary tool for the diagnosis of LEAD is its low sensitivity (61-73% appears too low for a screening method).This should be better stated in the conclusion paragraph.

-          Figure 1. Would it be cleared to present data as percentages (instead of absolute numbers)?

-          Figure 2: in the capture, “diagnosis” is repeated twice.

-          Line 176: “Comparison with other literature”. Please change to “Comparison with previous literature”

-          Please move the sentence in line 193-194 to the beginning of the paragraph

-          Line 202: “on” à “or”

Reviewer 2 Report

Danieluk et al. present a survey on a measurement tool that is first-line in the diagnosis of LEAD. Though limited by a small study sample, it gives a good overview on problems in access to simple diagnostic tools.

I have some points, which should be addressed, before this manuscript can be published:

.) Lines 41-46: Please also consider that PAD is most often a multisite disease:

In patients with PAD (lower extremity artery disease, ABI < 0.90),

•coronary artery disease is present in 25–70 %, •carotid artery stenosis (> 70 %) in 14–19 %, •and renal artery stenosis (> 75 %) in 10–23 %.   In patients with carotid artery disease •PAD is present in 18–22 %, •coronary artery disease in 7–16 %.

see and reference also PAD guidelines:

*) Frank U, Nikol S, Belch J, Boc V, Brodmann M, Carpentier PH, Chraim A, Canning C, Dimakakos E, Gottsäter A, Heiss C, Mazzolai L, Madaric J, Olinic DM, Pécsvárady Z, Poredoš P, Quéré I, Roztocil K, Stanek A, Vasic D, Visonà A, Wautrecht J-C, Bulvas M, Colgan M-P, Dorigo W, Houston G, Kahan T, Lawall H, Lindstedt I, Mahe G, Martini R, Pernod G, Przywara S, Righini M, Schlager O and Terlecki P. ESVM Guideline on peripheral arterial disease. Vasa. 2019;48:1-79.

*) Aboyans et al. ESC guidelines for PAD

.) in the introduction, please describe in a paragraph ABI measurements (how are they performed?), please discuss also the methodical differences (higher/ lower systolic blood pressure for calculation), see also

Lange et al. BMC Public Health 2007 7:147.

Fowkes FG et al.  JAMA. 2008 Jul 9;300(2):197-208.gives a good overview on different studies and the prognostic value of ABI measurements. Please discuss and reference the study.

.) Please also consider in the introduction/ discussion that stress tests increase the possibility of PAD detectin by 30% (ESVM guidelines).

.) in addition, it should be pointed out that an

ABI <0.40, ankle pressure <50 mmHg, toe pressure <30 mmHg, TcPO2 <30 mmHg define critical ischemia according to WIfI classification.

.) Line 66: perform ABI on their patients --> please re-phrase e.g. perform ABI measurements in the patients

.) Line 84:  please re-phrase: (..) Poland in the cities of Bialystock (University of Bialystock, Department of Family Medicine) and Krakow (Jagiellonian University, Department of Family Medicine)

.) Please include and discuss the getABI study: 

Vasa. 2002 Nov;31(4):241-8. doi: 10.1024/0301-1526.31.4.241.

.) In general, I would also highlight the shocking details that almost half of the trainees think that ABI is only moderately useful and about a quarter find it hard to assess or unnecessary. Moreover, >80% do not have access to ABI measurements in their workplace. Is there an explanation to the fact that trainees find the ABI moderately useful? Please discuss possible limitations of ABI (medial calcification, etc.).

I would also emphasize in the conclusions that training and awareness of young doctors should be promoted.
